# The Moderation of Perceived Comfort and Relations with Patients in the Relationship between Secure Workplace Attachment and Organizational Citizenship Behaviors in Elderly Facilities Staff

**DOI:** 10.3390/ijerph19020963

**Published:** 2022-01-15

**Authors:** Marcello Nonnis, Alessandro Lorenzo Mura, Fabrizio Scrima, Stefania Cuccu, Ferdinando Fornara

**Affiliations:** 1Department of Pedagogy, Psychology, Philosophy, University of Cagliari, 09123 Cagliari, Italy; cuccustefania@gmail.com (S.C.); ffornara@unica.it (F.F.); 2Department of Developmental and Social Psychology, Sapienza University of Rome, 00185 Rome, Italy; alessandrolorenzo.mura@uniroma1.it; 3Department of Psychology, Université de Rouen Normandie, 76130 Mont-Saint-Aignan, France; fabrizio.scrima@univ-rouen.fr

**Keywords:** health care workers, secure workplace attachment style, perceived comfort, relationship with patients, organizational citizenship behavior, residential facilities, occupational and organizational health, environmental health

## Abstract

This study focuses on caregivers who work in residential facilities (RFs) for the elderly, and specifically on their organizational citizenship behaviors (OCBs) in relation to their interaction respectively with the overall context (workplace attachment dimension), the spatial-physical environment (perceived environmental comfort), and the social environment (relationship with patients). A sample of health care workers (medical or health care specialists, nurses, and office employees, *n* = 129) compiled a self-report paper-pencil questionnaire, which included scales measuring the study variables. The research hypotheses included secure workplace attachment style as independent variable, OCBs as the dependent variable, and perceived comfort and relations with patients as moderators. Results showed that both secure workplace attachment and perceived comfort promote OCBs, but the latter counts especially as a compensation of an insecure workplace attachment. As expected, difficult relationships with patients hinder the relationship between secure workplace attachment style and OCBs. In sum, our study highlights the importance of the joint consideration of the psychological, social, and environmental dimensions for fostering positive behaviors in caregivers employed in elderly care settings.

## 1. Introduction

The topic of Active and Healthy Aging (HAA) from a psychological perspective [1,2,3,4] is particularly critical for the Italian population, which ranks first in Europe in terms of number of elderly people [5]. A response to this demographic configuration of Italian society and to the resulting demand for health is provided by Residential Facilities (RFs; “Residenze Sanitarie Assistenziali” (RSA) in Italian). These facilities may be public or private, are decentralized with respect to the National Health System (NHS) and play a complementary role in continuity with hospital facilities. In fact, their mission and objectives are related to the improvement of both health and socio-psychological conditions. In this regard, RFs are framed in a perspective not only of care and rehabilitation (i.e., loss reduction), but rather of health and well-being (i.e., gain achievement) in a general sense. This explains why in the last decade the RFs have spread in all the Italian territories [6,7].

RFs approach the issue of HAA from two complementary perspectives. The first concerns the structuring of therapy and rehabilitation pathways (mainly, but not exclusively) post hospitalization and post inpatient. These are aimed at restoring the dimensions of health or maintaining and enhancing the residual ones, in a bio-psycho-social perspective [8]. These are generally short, intensive, multidimensional, integrated, and personalized pathways, which aim to restore the health conditions of the elderly person and end with the reintegration into their social and territorial context [7,9]. The second perspective concerns the elderly with severe and/or chronic disabilities, who are often carriers of multiple and/or degenerative diseases and are therefore particularly fragile. Again, the goal is the maintenance and enhancement of residual health dimensions, but the elderly’s stay is significantly prolonged, and may become indefinite or definitive [10,11].

In the pursuit of HAA, particularly with the most frail elderly and those destined to long-term inpatient stays, RFs are supposed to be oriented toward achieving and maintaining high standards of response quality and healthcare humanization [12,13]. This perspective focuses on multiple dimensions, i.e., the physical health of the elderly (including monitoring and administration of care [9,14]), their emotions, and their culture, traditions, and habits [10]. In addition, the care of greenery (outside and inside the facilities [15]), aesthetics, comfort, and possibility of socialization [16] are of considerable importance. A further key facet of this orientation is represented by the occupational, organizational, and environmental health of staff working in RFs. In fact, it is necessary that operators are capable of caring work in their area of expertise, but also of cooperating within a multidisciplinary team that is open to discussion and improvement [17,18]. In addition, they should be able both to relate to the patient and his or her caregivers (often family members) with empathy, listening skills, emotional closeness, and affective exchange [19,20], and to know how to cope with critical relational situations, which can be marked by fear, suffering, anger, conflict, and misunderstanding [21], and not infrequently by despair and grief [22,23]. To promote job satisfaction in RFs staff, that is supposed to positively impact on the quality of care, again the characteristics of the spatial-physical environment are supposed to play an important role, in terms of functionality, welcoming appearance [24], and design humanization [16].

The importance that RFs have assumed in recent years, as a positive NHS response to the need for care and pursuit of HAA, is evidenced by the recent COVID-19 pandemic due to the SARS-CoV-2 virus [25]. In Italy the pandemic has highlighted the fragility of regional health systems and the importance of RFs. In fact, these facilities have vicariously served as hospital presidia, and their staff have cared for older and frail patients with a dedication and self-sacrifice that, in some cases, have put their health and safety at risk [26].

The psychological literature on HAA has primarily focused on the elderly people, whereas less attention has been devoted to caregivers as key resources for the promotion of elderly health in institutionalized settings [27,28]. The focus on RFs workers is motivated by their active role in providing support not only to the elderly residents, but also to their families [16]. In fact, a proper communication between the staff of healthcare environments and the older adults’ family can facilitate the involvement of family members in residents’ care, thus providing social support and reducing the level of anxiety [29]. Hence, the attention should be put on the quality of RFs caregivers’ experiences, since it can influence the residents’ health and well-being [30]. In this regard, there has been a lack of exploration about the connection between the relationship of RFs workers with their physical-spatial milieu and their behavioral responses. Following this line, our study focuses on organizational citizenship behaviors (OCB, [31,32]) of RFs caregivers, in relation to features covering different socio-psychological interfaces: (a) people-overall context interface-where the overall context includes both the spatial-physical and the social environment-represented by the workplace attachment style; (b) people-physical environment interface, represented by the perceived environmental comfort; (c) and finally, people-people interface, represented by the relationship between staff and patients. The following paragraphs report the literature ground which provides support to our choice to focus on such features.

### 1.1. Organizational Citizenship Behaviors in Healthcare Settings

According to Organ [31], organizational citizenship consists of the set of behaviors enacted by members of an organization, and that promote its efficiency and effectiveness. They are discretionary, not imposed by a contract, and do not involve formal rewards [32]. These behaviors are spontaneous and pro-social acts that go beyond the formally defined role and tasks. Thus, they are intra- and extra-role behaviors of fundamental importance for the effectiveness and efficiency of organizational processes and their improvement [33,34]. Organ [31] and other authors [35], have identified five dimensions of OCBs. The first is Conscientiousness, which refers to the special care a worker takes in performing his or her duties (e.g., strictly adhering to the job description). The second dimension is Civic Virtue, which refers to a strong sense of responsibility to the organization (for example, a willingness to offer advice to colleagues or to try to solve problems that arise at work). The third dimension, Sportsmanship, implies a sense of loyalty to one’s organization, which is manifested by emphasizing its best aspects and avoiding paying attention to the less positive ones. The fourth dimension, Altruism, expresses a willingness to help one’s co-workers (e.g., by helping new members or those with a higher workload). Finally, the fifth dimension, Courtesy, includes taking special care in establishing kind and cooperative relationships (e.g., trying to avoid relational conflicts between co-workers) [36]. In recent decades, literature on OCBs has highly expanded, with a focus both on OCBs’ nature and intensity, and on their antecedents and consequences [32,34,37,38]. A further issue concerns OCBs’ categorization, for which different criteria have been proposed. For example, Podsakoff et al. [39] distinguished OCBs with respect to two different worker orientations, i.e., OCBAs (behaviors oriented toward Affiliation to others, such as Altruism, Courtesy, and Interpersonal Help; or to the organization, such as Sportsmanship, Loyalty, and Compliance) and OCBCs (oriented to Challenge and Initiative, such as Civic Virtue and Making Constructive Suggestions).

Regarding healthcare organizations, several studies show the importance of the role played by OCBs. For example, Jahani et al. [40] showed that OCBs have a positive influence on organizational communication, promote organizational planning, improve interpersonal cooperation, develop a better organizational climate, and affect staff satisfaction, work life quality, service provision, job commitment, and financial output. Regarding the relationship between OCBs and human resources management practices in health care settings, Ranjhan and Mallick [41] showed the impact of OCBs on workers’ productivity, competitive advantage, and job performance in a study conducted with different job profiles of health care facilities (hospitals, nursing homes, and diagnostic centers). Another study, carried out by Saleh et al. [42], highlighted the influence of psychological capital and work engagement, on OCBs and organizational commitment.

In contrast, there are very few studies regarding OCBs of caregivers who work in RFs with the elderly. For instance, a study was led on caregivers of long term and community care facilities for the elderly by Ginsburg et al. [43], who highlighted the importance of work engagement and psychological empowerment on OCBs and turnover intention of caregivers, and how important these are for the quality of care provided by caregivers to the elderly people. In another study focusing on the same target population (i.e., long term care and community care operators), Perreira and colleagues [44] found that organizationally oriented OCBs are related to psychological empowerment, job satisfaction, and intention to remain in the same work setting.

### 1.2. Workplace Attachment Styles

The development of positive job-related responses in workers can be also promoted by a positive workplace attachment, which is a specific pattern included in the broader construct of place attachment, defined as an individual’s bond (mainly of affective nature) to a meaningful place [45]. Place attachment is conceived as composed of two dimensions, i.e., place dependence [46], that is a form of functional attachment to a specific place, and place identity [47], that refers to a substructure of the self-including cognitions, emotions, and behavioral tendencies toward the physical environment [48]. The interest for the place attachment topic is witnessed by the noteworthy number of studies that have addressed this pattern as implied in various outcomes, such as individual well-being [49], social well-being [50], customer behavior [51], environmentally responsible behavior [52] or satisfaction [53], coping behaviors in environmental disasters [54]. A recent theoretical proposal conceives place attachment in the light of Bowlby’s classical theory of attachment [55]. In fact, according to Scannell [56], places can also act as a “safe haven” for individuals. In that case the specific places will have the role of protecting the individual, solving problems and emotional relief. According to Morgan [57], children develop internal operational models (IWMs) in relation to their experiences with places. Such IWMs would therefore be formed during childhood as a function of the relationships between the child and their home [58] and the characteristics of these IWMs will influence future relationships between individuals and other specific places [59] such as the city [60], the neighborhood [61], or the workplace [62]. For example, Scrima, et al. [63] found that it is possible to identify attachment styles to the workplace. Taking up the model of Bartholomew and Horovitz [64], Scrima et al. [63] argue that place attachment styles are a function of the thoughts of Self and thoughts of Place. Positive thoughts of Self and Place refer to a secure attachment style, positive thoughts of Self and negative thoughts of Place are associated with an avoidant attachment style, negative thoughts of Self and positive thoughts of Place are associated with a preoccupied attachment style, and negative thoughts of Self and Place characterize the disorganized attachment style. In particular, secure attachment style develops when a specific place satisfies the needs of its user [65].

Thus, a workplace that meets the employee’s needs should allow the employee to develop a secure attachment style with the workplace. The development of a secure attachment style to the workplace seems to have numerous beneficial aspects. For example, a secure style promotes exploration, since this allows individuals to walk away without feeling discomfort [66]. According to Stancu et al. [67], a secure place attachment is a buffer against psychological distress and improves constructive and adaptive strategies in risk coping. To the best of our knowledge, there have been no studies so far focusing on workplace attachment styles in health care and RFs settings.

### 1.3. The Physical-Spatial Comfort of Caregivers in Healthcare Settings

The literature on the effect of the workplace spatial-physical features on office employees’ well-being and behaviors has evidenced that design-related dimensions such as noise, lighting, temperature, and indoor air quality conditions can play a role in employees’ satisfaction, attitudes, behaviors, and performance (e.g., [68]). The lack of quality of the workplace design can in fact elicit a low level of work satisfaction and job productivity, poor health, and high mental fatigue [69]. About the healthcare settings, the influence of spatial-physical features of the workplace in caregivers’ outcomes has been evidenced by the extended literature review by Ulrich and colleagues [70], who showed that different aspects of the built environment can affect the degree of staff injuries, stress, work effectiveness, and satisfaction. The importance of considering the impact of healthcare workers’ perceived comfort on their well-being and behaviors is explicitly remarked in the architectural humanization approach to the healthcare environments [71,72,73]. According to this approach, more humanized healthcare settings elicit a higher perceived quality of design attributes that should be provided in order to satisfy users’ needs, including sense of welcome, orientation, and comfort, both spatial (i.e., office furniture, layout, and spatial characteristics) and sensorial (in visual terms, i.e., adequate lighting and panoramic views; auditory terms, i.e., avoidance of annoying noises; and climatic terms, i.e., adequacy of temperature and humidity [74]).

### 1.4. The Relationships between Patients and Caregivers in Healthcare Settings

In the perspective of the humanization of care settings, a patient-centered approach, takes on particular importance. This orientation is focused on the involvement of the patient as an active and autonomous subject (with physical, relational, and emotional needs [75]), and on an adequate relationship between the patient and the caregiver [76]. Regarding this last aspect, several authors highlight how empathy is a supportive factor in the process of care, since it promotes patient’s compliance and treatment positive outcomes, and facilitates the well-being of health care providers themselves (e.g., [77,78]). Other important antecedents of an adequate caregiver-patient relationship are the caregiver professionals’ ability to actively listen [79] and communication skills in terms of assertiveness [80,81]. With regard to RFs, some scholars have emphasized the importance of the relationship between health care providers and the residents (e.g., [82]), whilst another study highlighted the operators’ professional skills involved in the care of the elderly, such as bonding and connection, tenderness and closeness, non-infantilization and respect [83]. Such skills can be considered as positive extra-role behaviors, which are necessary for the achievement and maintenance of an adequate level of relational humanization in these health care settings.

### 1.5. Objective and Hypotheses

The aim of this research is to verify the role of secure workplace attachment style (tapping people-overall context interface), perceived comfort (concerning people-physical environment interface) and relationship staff-patients (regarding people-people interface) in the RFs workers’ OCBs.

The relationship between workplace attachment and OCBs (or similar organizational behaviors), is still substantially unexplored, with a few exceptions. For instance, Desivilya [84] showed that the secure attachment style is an antecedent of pro-social and positive behaviors in organizations, and among them OCBs. Harms et al. [85] found that employees’ avoidant attachment style toward their bosses negatively affects their OCBs directed at other individuals. Finally, Reizer [86] showed that Self-compassion mediates the negative influence of avoidant and anxious attachment styles on OCBs. It is to remark that none of these studies were carried out in health care settings or RFs. Thus, we hypothesized that:

**Hypothesis** **1** **(H1).**
*Secure workplace attachment is positively related to OCBs, both in terms of overall factors and concerning the distinct OCBs components (i.e., Altruism, Conscientiousness, and Civic Virtue).*


Studies on the influence of physical-spatial comfort and organizational behaviors such as OCBs are generally lacking. Even though it has been shown that office physical features such as noise, lighting conditions, temperature, and indoor air quality impact the attitudes, behaviors, productivity, and satisfaction of employees [68], and that the low quality of workplace design can elicit lower work satisfaction, lower job productivity, poorer health, and higher mental fatigue [69], less attention has been placed on perceived design quality in relation to workers’ behaviors and well-being. Among those addressing these concerns, Zoghbi-Manrique-de-Lara et al. [87], evidenced the negative influence of crowded physical workspaces with low privacy on organizational citizenship behaviors in favor of others (OCB-I) in a study with workers in open-space facilities. Carter and colleagues [88] found that satisfaction with physical workspace and perceived quality of the physical workplace are drivers of OCBs. Scrima et al. [89] detected a significant impact of satisfaction toward the workplace design on exhaustion, i.e., a subdimension of burnout.

We expected both a main effect of perceived spatial-physical comfort on OCBs and an amplification effect of the relationship between secure workplace attachment and OCBs:

**Hypothesis** **2** **(H2).**
*Perceived spatial-physical comfort is positively related to OCBs, both in terms of overall factors and concerning the distinct OCBs components.*


**Hypothesis** **3** **(H3).**
*Perceived spatial-physical comfort moderates the relationship between secure workplace attachment and OCBs, both in terms of overall factors and concerning the distinct OCBs components.*


Regarding the bond of staff-patient relationships with OCBs or similar behaviors, Jarfarpanah et al. [90], found a connection between patient safety culture and OCBs Altruism, Civic Virtue, and Courtesy in nurses. Wibowo and colleagues [91], showed that OCBs in hospital settings facilitate better performance in caring work with patients. Perreira et al. [92] showed that caregivers’ affective commitment to patients positively influences their OCBs, Finally, Lavee and Pindek [93] found that commitment to care work can lead caregivers to exhaustion (i.e., an outcome that is opposite to OCBs) under conditions of high job demand in health, education, and welfare service providers.

As well as in the case of perceived spatial-physical comfort, we hypothesized both a main negative effect of difficult staff-patient relationships on OCBs and a buffer effect of the link between secure workplace attachment and OCBs. Thus, we expected that:

**Hypothesis** **4** **(H4).**
*Difficult staff-patient relationships are negatively related to OCBs, both in terms of overall factors and concerning the distinct OCBs components.*


**Hypothesis** **5** **(H5).**
*Difficult staff-patient relationships moderate the relationship between secure workplace attachment and OCBs, both in terms of overall factors and concerning the distinct OCBs components.*


## 2. Materials and Methods

### 2.1. Research Protocol

Data were collected between October 2019 and February 2020 through a self-report paper-pencil questionnaire administered to operators of three RFs in the Autonomous Region of Sardinia (Italy). The sample was of convenience and those who participated in the research volunteered (without receiving any compensation). Although the sample is not representative of the population of RFs caregivers, we respected the structure of job roles found in RFs. A total of 164 research questionnaires were distributed, of which 129 were returned (response rate = 78.65%).

### 2.2. Participants

The final sample of 129 practitioners consists mostly of females (77.51%, *n* = 100), whereas males are about one quarter (22.49%, *n* = 29). Approximately half of the participants (53.48%, *n* = 69) are health care workers (in Italian “Operatore Socio Sanitario, (OSS)”); 22.48% (*n* = 29) are medical or health care specialists; 13.95% (*n* = 18) are nurses; and about the remaining 10% are office employees (*n* = 13). Most are either graduates (37.98%, *n* = 49) or high school graduates (44.96%, *n* = 58) and have an average length of service of 11.18 years (SD = 6.95). Finally, about half are between the ages of 31 and 45 (51.93%, *n* = 67) and about one-third are between the ages of 46 and 60 (31.08%, *n* = 40).

### 2.3. Instruments

The questionnaire contains the following measurement scales.

*Organizational Citizenship Behavior* (OCB): We used the brief version (15 items) of the Organizational Citizenship Behavior Scale [35] in its Italian validation [94]. The Principal Component Analysis performed on the scale confirmed its 3-component factorial structure, including Altruism (5 items, Alpha = 0.89, e.g., “I help others who have a heavy workload”), Conscientiousness (3 items, Alpha = 0.73, e.g., “I do my job without constant requests from my boss”) and Civic Virtue (4 items, Alpha = 0.82, e.g., “I keep abreast of changes in the organization”). We used both these three specific components and the overall OCB (15 items, Alpha = 0.89) for the inferential analyses.

*Secure workplace attachment style*: from the Workplace Attachment Scale (Echelle d’Attachement au lieu de travail–EALT, [62]), recently validated in Italian by Scrima [95], the five items included in the secure workplace attachment style (Alpha = 0.84, e.g., “I am attached to my workplace”) were used;

*Perceived physical-spatial comfort*: A 16-item scale adapted from the short version of the Perceived Healthcare Environment Quality Indicators (IUOP, [73,96]) was used to assess the perceived physical-spatial comfort (Alpha = 0.87, e.g., “In this in-patient/waiting area furnishings are in good condition”);

*Difficult relationship with patients*: for the assessment of the relationship between RFs workers and patients, the five items of the component “ambiguous customer expectations” of the Customer-Related Social Stressors scale by Dormann and Zapf [97] were used (Alpha = 0.88, e.g., “Customers’ wishes are often contradictory”).

For all these measures, the response scales consisted of a 7-step Likert-type scale (from 1 = “completely disagree” to 7 = “fully agree”).

Socio-demographic data (sex, age, and length of service expressed in range, occupational role, and level of education) were also collected.

### 2.4. Data Analysis

The mono-factorial structure of the scales measuring secure workplace attachment style, perceived spatial-physical comfort, and difficult relationship with patients was verified through a Principal Component Analysis (on IBM SPSS 26). As concerns the factorial structure of the OCBs scale (see Appendix A), which has been conceived and then tested as multi-factorial in previous literature (e.g., [31]), a confirmatory factor analysis (CFA) was run (on JASP 0.14). The model including the 3-factor structure and the second-order overall factor was confirmed, showing acceptable fit indices (CFI = 0.96; TLI = 0.95; RMSA = 0.07 [90% CI = 0.04, 0.10]; SRNR = 0.04; χ^2^/df = 1.6).

To test the research hypotheses, model 2 (Figure 1a; Figure 1b) of the process macro (on IBM SPSS 26) was used [98], including a secure workplace attachment style as the independent variable (IV), overall OCB and the three specific OCB subdimensions (Altruism, Conscientiousness, and Civic Virtue) as dependent variables (DVs), and perceived comfort and difficult relationship with patients as moderators.

The simple slope analysis allowed us to interpret the moderation effects of perceived comfort and difficult relationship with patients. Sex, age, education level, and marital status were inserted as covariates.

### 2.5. Ethical Issues

The research was authorized by the Ethics Committee of the University of Cagliari (approval number 73624 dated 30 March 2021). It was conducted in full compliance with the Ethical Principles of Psychologists and Code of Conduct of the American Psychological Association (APA), integrated into the Associazione Italiana Psicologia (AIP) code of ethics. Furthermore, the study did not address any sensitive topics and was carried out via procedures for informed and consenting adults. Lastly, in accordance with the Italian privacy law, the research ensured the anonymity and privacy of all participants.

## 3. Results

### 3.1. Preliminary Analyses

The results of the factor analysis on the multidimensional scales suggested the use of a single-factor measure for the scales of perceived comfort, secure workplace attachment style, and problematic relationships with patients; and a 3-factorial structure for the OCB scale (Altruism, Conscientiousness, and Civic Virtue).

Table 1 shows the descriptive and correlational analyses performed on the aggregate variables computed after the verification of factorial solution and reliability.

Regarding the correlations between socio-demographic variables and the variables under study, age (r = −0.26, *p* < 0.01) and marital status (r = −0.21, *p* < 0.01) are negatively associated with Altruism, while education level is positively correlated with perceived comfort (r = 0.19, *p* < 0.05). As expected, overall OCB correlates positively with secure workplace attachment style (r = 0.34, *p* < 0.00001). Specifically, secure workplace attachment correlates with Altruism (r = 0.30, *p* < 0.0001) and with Civic Virtue (r = 0.36, *p* < 0.0001), but not with Conscientiousness (r = 0.17, *p* = n.s.). Furthermore, overall OCB correlates positively with perceived comfort (r = 0.42, *p* < 0.0001). The three dimensions of OCB, Altruism, Conscientiousness and Civic Virtue, are positively associated (*p* < 0.0001) with perceived comfort, showing coefficients of 0.35, 0.33, and 0.30 respectively. It is also interesting to note that no OCB dimension is associated with difficult relationship with patient. Finally, secure workplace attachment correlates positively with perceived comfort (r = 0.39, *p* < 0.0001)

### 3.2. Hypotheses Testing

Four independent models were tested to verify the five hypotheses: one for the overall OCB and the other three for its components (Altruism, Conscientiousness, and Civic Virtue). As expected (H1), secure workplace attachment style is positively related with overall OCB (B = 0.13, *p* < 0.05) (Table 2), Altruism (B = 0.13, *p* < 0.05) (Table 3) and Civic Virtue (B = 0.26, *p* < 0.01) (Table 4).

Contrary to what has been hypothesized, secure workplace attachment style is not positively related with Conscientiousness in the workplace (B = 0.04, *p* = n.s.) (Table 5).

A statistically significant relationship emerged from the test of the direct effect of perceived comfort on the OCB (H2): the perceived comfort was positively related with overall OCB (B = 0.26, *p* < 0.01) (Table 2), Altruism (B = 0.27, *p* < 0.01) (Table 3) and Conscientiousness (B = 0.28, *p* < 0.01) (Table 5) but not with Civic Virtue (B = 0.20, *p* = n.s.) (Table 4).

Unlike what was expected (H3) only the interaction between secure WA and perceived comfort on overall OCB was statistically significant (B = −0.10, *p* < 0.05) (Table 2; Figure 2, Simple slope a).

No statistically significant interaction between secure WA and perceived comfort emerged on the other dependent variables, Altruism (B = −0.08, *p* = n.s.) (Table 3), Conscientiousness (B = −0.11, *p* = n.s.) (Table 4) and Civic Virtue (B = −0.10, *p* = n.s.) (Table 5).

Likewise, the results relating to H4 do not highlight any direct effect of difficult relationship with patient with overall OCB (B = 0.07, *p* = n.s.) (Table 2), nor with the components of Altruism (B = 0.09, *p* = n.s.) (Table 3), Civic Virtue (B = 0.03, *p* = n.s.) (Table 4) and Conscientiousness (B = 0.04, *p* = n.s.) (Table 5).

The results referring to the interaction between secure WA and difficult relationship with patients fully support the hypotheses (H5): as emerged from the simple slope analysis, the interaction is negatively associated with overall OCB (B = −0.10, *p* < 0.01) (Table 2; Figure 2, Simple slope b), Altruism (B = −0.08, *p* < 0.05) (Table 3; Figure 2, Simple slope c), Conscientiousness (B = −0.08, *p* < 0.01) (Table 5; Figure 2, Simple slope d), and Civic Virtue (B = −0.15, *p* < 0.01) (Table 4; Figure 2, Simple slope e).

## 4. Discussion

Regarding H1, the emergence of the expected positive relationship between secure attachment to the workplace and organizational citizenship behaviors substantially confirmed what has been shown in other studies (e.g., [84]), especially with reference to altruistic and virtuous behavior towards the organization. However, previous evidence concerns indeed a different source of the secure attachment, that is the job itself rather than the workplace. Having said that, our results provide further proof about the importance of developing a secure attachment at work to promote positive behaviors.

About H2, it is substantially confirmed. In fact, in line with Carter et al. [88], caregivers’ perceived spatial-physical comfort in the work environment increases the likelihood of performing OCBs. In particular, the relationships emerged between perceived comfort and altruistic behaviors, this is in line with the negative effects of low comfort on workers’ altruism found in the study by Zoghbi-Manrique-de-Lara et al. [87].

Concerning H3, a pattern opposite to the one expected was found. In fact, perceived comfort played a buffering rather than an amplifying role in the moderation of the relationship between secure workplace attachment style and OCBs. It is to note that this was found only for the overall OCBs and not for the specific components. This counterintuitive finding, which requires further investigation, can be explained in terms of compensation provided by a comfortable environment for those caregivers with an insecure attachment to the workplace, whereas this environmental perception counts less in case of secure workplace attachment. In other words, the fact that operators perceive a condition of physical-spatial comfort overshadows (and depowers) the need for them to “draw on” their secure workplace attachment style as a motivational dimension capable of promoting their OCBs.

Regarding H4, contrary to what was expected, the difficulties faced by staff in their relationships with patients do not negatively affect their implementation of OCBs. A possible explanation for this lack of negative incidence can lie in the awareness that RFs operators have of the particular conditions, non-self-sufficiency, and often mental pathology of their patients (often suffering from senile dementia or similar disabling mental pathologies related to aging). This awareness can lead practitioners to a high level of tolerance and compassionate forbearance of the relational difficulties that may arise with these type of patients (see e.g., [79]), without negatively affecting their propensity to implement OCBs.

Finally, H5 is confirmed, since the difficulties in the relationships with patients adversely affect the relationship between caregivers’ secure workplace attachment and their OCBs (both general OCBs and the specific components, i.e., Altruism, Civic Virtue, and Conscientiousness). This result suggests that it is also important to consider the relational dimension of the secure workplace attachment style, both towards the social work environment (as found by Afshar et al. [50]) and towards users (consistently with findings of Alexandris et al. [51]). In this sense, it is therefore plausible that problematic relationships with users may depower the ability of the secure workplace attachment style to foster organizational citizenship behaviors.

This study has some limitations. The first one concerns the limited number of participants, due to the specific target population (i.e., staff of residential facilities who work with the elderly), that is not easy to reach and investigate for its special characteristics (including the difficulty to obtain access to elderly facilities for survey aims). In addition, participants were identified based on their availability, without a sampling method that would allow us representativeness of these workers and generalizability of our results. However, the sample number is good enough for ensuring the reliability of our results, given the low number of variables that we included in the tested moderation model. A second limitation is given by the lack of objective parameters of the design features, since we relied only on the staff assessment of their spatial-physical environment. Thus, future research should fill this gap by considering also objective indicators of the workplace environment, such as spatial configuration, lighting conditions, indoor and outdoor views, furniture, etc., to compare with users’ perceived comfort. Finally, it would be interesting to confirm our speculations about why the independent or moderating variables in our study did not affect OCBs (as for H4) or affected them in an unexpected way (i.e., the counterintuitive results related to H3) through the realization of a qualitative exploration (e.g., focus groups of semi-structured interviews) of practitioners’ views.

Despite these limitations, the outcomes of our study emphasize the importance of the joint consideration, for the implementation of OCBs in the RFs, and more generally in residential care settings, of dimensions tapping aspects concerning different socio-psychological interfaces, i.e., covering the overall context (including both the spatial-physical and the social environment), mirrored by the workplace attachment style; the spatial-physical environment, represented by the perceived environmental comfort; and finally the people facet, reflected by the relationships between staff and patients. In other words, this study highlights that, in order to promote positive caregiver behaviors (OCBs) towards older adults and the healthcare organization, it is crucial to jointly consider place-based dimensions, such as the workplace attachment style and the perceived environmental comfort, as well as the socio-relational dimension represented by the helping relationship towards patients.

## 5. Conclusions

Our study lies within the theoretical framework of positive organizational behaviors (POBs) enacted by individuals, teams, organizations, and the health network as a whole [8,99], in the light of the promotion of healthcare humanization for the older adults [82,83], consistently with a healthy and active aging perspective [1,2]. In this view, caregivers’ OCBs can be considered as desirable outcomes promoting HAA. From a practical point of view, the development of a secure attachment to the workplace can be fostered through listening to, engaging with, and meeting the needs of caregivers in care settings [65]. This should encourage them to be proactive in exploring innovative ways to benefit the organization and the patients (and thus enact organizational citizenship behaviors) and promote their work well-being [67]. Furthermore, the achievement or maintenance of a high level of physical-spatial comfort, according to an architectural humanization approach to the healthcare environments [72], requires the participation of the actors involved in the monitoring, evaluation, and improvement of the healthcare environments [74]. In addition, the quality of the staff-patients relationship in these settings from a patient-centered perspective [75] is particularly important for the elderly population. Therefore, it is appropriate that caregivers carry out periodically and systematically learning update activities for improving their listening skills [79], empathy [77,78], and assertiveness [81]. It is also important that caregivers benefit from supervision for avoiding the risks of exhaustion [89] or compassion fatigue [23] resulting from a poor helping relationship with patients that, for its emotional charge, can jeopardize their work well-being and hinder their commitment to active and healthy aging of the elderly.

## Figures and Tables

**Figure 1 ijerph-19-00963-f001:**
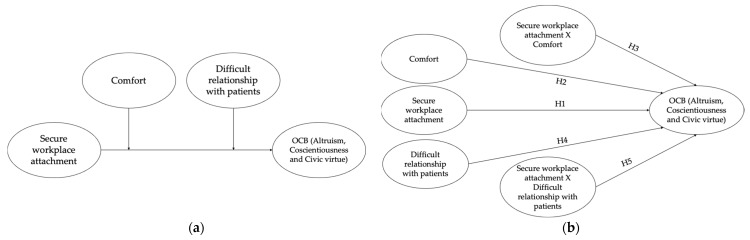
(**a**) Conceptual model of the tested model. (**b**) Diagram of the tested model.

**Figure 2 ijerph-19-00963-f002:**
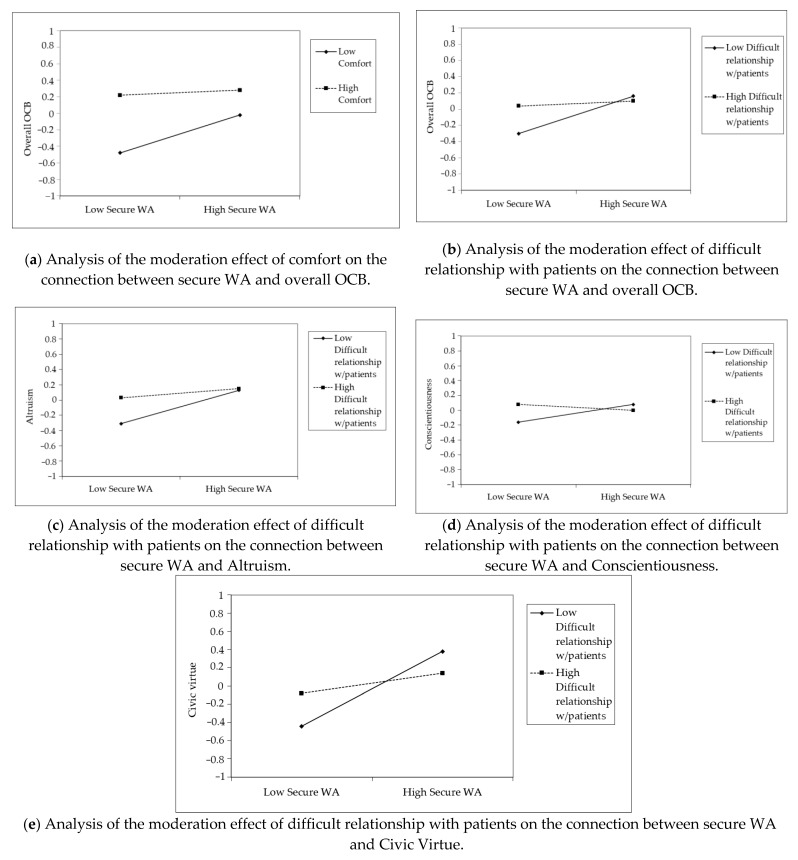
Simple slope analysis of the moderation effect of perceived comfort and difficult relationship with patients, on the connection between secure WA and OCBs (Overall, Altruism, Conscientiousness, and Civic Virtue).

**Table 1 ijerph-19-00963-t001:** Means, SDs and bivariate correlations of the study variables, and alpha in diagonal.

		Mean	SD	1	2	3	4	5	6	7	8	9	10	11
1	Sex (1 = M, −1 = F)	-	-	-										
2	Age	-	-	0.05	-									
3	Marital status	-	-	−0.10	0.36 ***	-								
4	Education level	-	-	0.03	−0.35 ***	−0.25 **	-							
5	Secure workplace attachment	4.00	1.39	−0.03	−0.01	0.02	0.02	-						
6	Comfort	4.79	0.90	0.05	−0.02	−0.05	0.19 *	0.39 ***	-					
7	Difficult relationship with patients	4.01	1.39	−0.03	−0.04	−0.07	−0.04	−0.10	−0.11	-				
8	OCB	5.72	0.84	−0.11	−0.12	−0.12	0.08	0.34 ***	0.42 ***	0.11	-			
9	Altruism	5.92	0.95	−0.10	−0.26 ***	−0.21 *	0.06	0.30 ***	0.35 ***	0.13	0.84 ***	-		
10	Conscientiousness	5.84	0.98	−0.13	−0.06	−0.05	0.05	0.17	0.33 ***	0.06	0.74 ***	0.52 ***	-	
11	Civic Virtue	5.21	1.29	−0.05	0.07	−0.02	0.07	0.36 ***	0.30 ***	0.02	0.81 ***	0.51 ***	0.43 ***	-

Note: *n* = 129; *****: *p* < 0.05; ******: *p* < 0.01; ***: *p* < 0.001; SD: standard deviation; OCB: organizational citizenship behavior.

**Table 2 ijerph-19-00963-t002:** Moderation model with overall OCB as DV.

	B	SE	t	95% LLCI	95% ULCI
*Covariate*					
Sex	−0.15	0.08	−2.00 *	−0.30	−0.01
Age	−0.06	0.10	−0.60	−0.25	0.14
Marital status	−0.05	0.06	−0.81	−0.17	0.07
Education level	−0.09	0.09	−0.96	−0.27	0.09
*Independent Variable*					
Secure WA	0.13	0.05	2.52 **	0.03	0.22
*Moderator*					
Comfort	0.26	0.08	3.22 **	0.10	0.42
Difficult relationship with patients	0.07	0.05	1.47	−0.02	0.16
*Interaction*					
Secure WA x Comfort	−0.10	0.04	−2.30 *	−0.18	−0.01
SecureWAxDifficultrelationshipwithpatients	−0.10	0.03	−3.32 **	−0.16	−0.04

Note: Dependent variable = OCB; R^2^ = 0.35; *****: *p* < 0.05, ******: *p* < 0.01 R^2^ ChangeSecureWAxComfort = 0.03 (F_(1, 121)_ = 5.27, *p* < 0.05); R^2^ ChangeSecureWAxDifficultRelationWithPatients = 0.06 (F_(1, 121)_ = 11.00, *p* < 0.01); SE: standard error; LLCI: lower level of confidence interval; ULCI: upper level of confidence interval.

**Table 3 ijerph-19-00963-t003:** Moderation model with secure WA as IV and Altruism as DV.

	B	SE	t	95% LLCI	95% ULCI
*Covariate*					
Sex	−0.14	0.09	−1.61	−0.32	0.03
Age	−0.27	0.11	−2.34 *	−0.50	−0.05
Marital status	−0.11	0.07	−1.53	−0.25	0.03
Education level	−0.18	.11	−1.70	−0.39	0.03
*Independent Variable*					
Secure WA	0.13	0.06	2.31 *	0.02	0.25
*Moderator*					
Comfort	0.27	0.09	2.84 **	0.08	0.46
Difficult relationship with patients	0.09	0.05	1.71	−0.01	0.20
*Interaction*					
Secure WA x Comfort	−0.08	0.05	−1.60	−0.18	0.02
SecureWAxDifficultrelationshipwithpatients	−0.08	0.04	−2.31 *	−0.15	−0.01

Note: Dependent variable = Altruism; R^2^ = 0.32; *****: *p* < 0.05, ******: *p* < 0.01; R^2^ ChangeSecureWAxComfort = 0.02 (F_(1, 121)_ = 2.55, *p* = 0.11); R^2^ ChangeSecureWAxDifficultRelationWithPatients = 0.03 (F_(1, 121)_ = 5.34, *p* < 0.05); SE: standard error; LLCI: lower level of confidence interval; ULCI: upper level of confidence interval.

**Table 4 ijerph-19-00963-t004:** Moderation model with secure WA as IV and Civic Virtue as DV.

	B	SE	t	95% LLCI	95% ULCI
*Covariate*					
Sex	−0.15	0.13	−1.20	−0.41	0.10
Age	0.26	0.16	1.57	−0.07	0.58
Marital status	−0.03	0.10	−0.25	−0.23	0.18
Education level					
*Independent Variable*					
Secure WA	0.26	0.08	3.15 **	0.10	0.43
*Moderator*					
Comfort	0.20	0.14	1.52	−0.07	0.46
Difficult relationship with patients	0.03	0.08	0.41	−0.12	0.18
*Interaction*					
Secure WA x Comfort	−0.10	0.07	−1.34	−0.24	0.05
SecureWAxDifficultrelationshipwithpatients	−0.15	0.05	−2.82 **	−0.25	−0.04

Note: Dependent variable = Civic Virtue; R^2^ = 0.24; ******: *p* < 0.01; R^2^ ChangeSecureWAxComfort = 0.01 (F_(1, 121)_ = 1.80, *p* = 0.18); R^2^ ChangeSecureWAxDifficultRelationWithPatients = 0.06 (F_(1, 121)_ = 7.92, *p* < 0.01); SE: standard error; LLCI: lower level of confidence interval; ULCI: upper level of confidence interval.

**Table 5 ijerph-19-00963-t005:** Moderation model with secure WA as IV and Conscientiousness as DV.

	B	SE	t	95% LLCI	95% ULCI
*Covariate*					
Sex	−0.19	0.10	−1.87	−0.38	0.01
Age	0.02	0.13	−0.14	−0.27	0.24
Marital status	−0.01	0.08	−0.13	−0.17	0.15
Education level	−0.07	0.12	−0.57	−0.30	0.17
*Independent Variable*					
Secure WA	0.04	0.07	0.58	−0.09	0.17
*Moderator*					
Comfort	0.28	0.11	2.63 **	0.07	0.49
Difficult relationship with patients	0.04	0.06	0.63	−0.08	0.16
*Interaction*					
Secure WA x Comfort	−0.11	0.06	−1.92	−0.22	0.01
SecureWAxDifficultrelationshipwithpatients	−0.08	0.04	−2.09 *	−0.16	−0.01

Note: Dependent variable = Conscientiousness; R^2^ = 0.19; *****: *p* < 0.05, ******: *p* < 0.01; R^2^ ChangeSecureWAxComfort = 0.03 (F_(1, 121)_ = 3.68, *p* = 0.06); R^2^ ChangeSecureWAxDifficultRelationWithPatients = 0.03 (F_(1, 121)_ = 4.36, *p* < 0.05). SE: standard error; LLCI: lower level of confidence interval; ULCI: upper level of confidence interval.

## Data Availability

Upon reasonable request, data used and analyzed during the current study are available from the corresponding author.

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
