# Peer review of "The Moderation of Perceived Comfort and Relations with Patients in the Relationship between Secure Workplace Attachment and Organizational Citizenship Behaviors in Elderly Facilities Staff"

_ijerph, 2022, doi:10.3390/ijerph19020963_

Round 1

Reviewer 1 Report

The topic is very interesting and impressive.I believe the research will have contributions to the existing literature. While, still some problems should be paid attention to as follows. 

Methods

This research is not enough to introduce the source of the data, collection methods, and sampling methods. Although the article has some designs, as a scientific research, a separate section should be made to explain the data collection process and the evaluation of the representativeness of the data very clearly.

Measurement

1.The variables used by the author, whether dependent variables or independent variables(Organizational Citizenship Behavior,Secure  workplace  attachment  style Perceived physical-spatial comfort. Difficult relationship with patients), are measured by various scales, but the authors do not show them very clearly. The results of exploratory factor analysis should be shownin the texts,or at least be included as attached materials

2.There are multiple figures of moderating effects in the article, and the author can combine them into one figure yo save the space.

Results

1.In the result analysis part, the authors should first describe the overall situation of OCB as a whole, as well as the situation of each sub-dimension(Altruism,Conscientiousness and Civic Virtue), and interact with the respective variables as much as possible, because the described results usually reflect those variables that may affect the dependent variables. Although the authors show the correlation coefficient matrix of the dependent variable and each main independent variable, it is still not enough.

2. In the model results of this article, the only control variables are gender and age, but probably education level and marital status are likely to be related to the dependent variables, so they also need to be controlled. It is recommended to add these two control variables to the model.

Author Response

Dear Reviewer #1, please see our responses to all your comments, in the attachment.

Kind Regards

The Authors

Reviewer 2 Report

Dear Authors, I have read your manuscript with interest.

The current manuscript titled: "The Moderation of Perceived Comfort and Relations with Patients in the Relationship between Secure Workplace Attachment and Organizational Citizenship Behaviors in Elderly Facilities Staff" represents an important analysis of evolving field of Psychology.

The title reflects the manuscript content and helps the reader navigate the article essence.

The abstract contains all the necessary information in a concise form.

The introduction section is clear and easy to read. It provides the basic overview of the current problem, well documented.

The author described in detail the methods used, patient group, method of data extraction and the statistical methods used to process the presented data.

The result section is well written and detailed. For a better understanding of results, the Authors has attached tables and figures.

The discussion section is within the context of the results section and compares the results to other studies. There is sufficient references number, and the authors cite appropriate number of previous studies in the area.

In my opinion, these are the adjustments which should be made to increase the value of your manuscript:

  1. Line 74: add please abbreviation for COVID-19.
  2. Figures 1a and 1b are difficult to trace due to their small font and structure. Revise the font size and font to fit the general text.
  3. The manuscript contains many punctuation errors, please revise the text (e.g., lines 129, 279, etc.).
  4. In tables 1-5, add please the abbreviations for shortcut words (e.g., SD, OCB, LLCI, etc.).
  5. The conclusion section is well written, however, I advise moving some of the information to the Discussions section so that the Conclusions section to be more synthesized and specific. Moreover, it is not clear the practical relevance of this study. Please, clarify this point.

Good luck!

Author Response

Dear Reviewer #2, please see our responses to all your comments, in the attachment.

Kind Regards

The Authors

Reviewer 3 Report

Very well written. Some minor grammar, spelling, and punctuation errors, run the manuscript through Grammarly or another platform to catch these issues. Line 34 was flagged as plagiarism along with other lines.

General Concept comments: (weaknesses, method inaccuracies)

Grammar, spelling, and punctuation were noted as slight weaknesses, easily remedied with the appropriate editors or online platforms.

Method presentation was very complete and easy to follow. Not sure I noticed any information about the generalizability of study results.

Specific Comments: (line numbers, table numbers, figure numbers, pointing out inaccuracies, etc.)

Line 15: nurses and administrative ????? (N=129) complied…

Line 56-61: is it possible to combine some elements of this long list?

Line 80+, 86: Clearly identified problem section, great!

Line 129: could not find the meaning of HRM practices (what does HRM stand for)?

Line 135, 178, 222 +: Gap in evidence-literature identified several times, great!

Line 251: H2 consistency with other hypotheses: italics

Line 282: spell out the fraction ¼

Line 327: Figure 1.a. 1.b.: enlarge bubble legends, hard to read

Line 350: Table 1: Row 9=Civic virtue, column 4: reads .030***, should this be 0.30*** (leading zero)

Author Response

Dear Reviewer #3, please see our responses to all your comments, in the attachment.

Kind Regards

The Authors

Round 2

Reviewer 1 Report

some revisions are made,  it will be better if the authors could  taken consideration the following advice enough .

1.The author should provide the results of the factor analysis, and report the results and the screeplot instead of just enumerating scale items, these results must be included in maintext or as supplementary materials which will show the scientific of the analysis.

2.Put all the figures together to display to save the valuable space.

Author Response

Dear Reviewer 1, thank you for your comments.  Please find attached our responses to all your suggestions.

The Authors
